# Andrographolide and Its 14-Aryloxy Analogues Inhibit Zika and Dengue Virus Infection

**DOI:** 10.3390/molecules25215037

**Published:** 2020-10-30

**Authors:** Feng Li, Wipaporn Khanom, Xia Sun, Atchara Paemanee, Sittiruk Roytrakul, Decai Wang, Duncan R. Smith, Guo-Chun Zhou

**Affiliations:** 1School of Pharmaceutical Sciences, Nanjing Tech University, Nanjing 211816, China; fengli9203@163.com (F.L.); sunxia0107@126.com (X.S.); dcwang998@126.com (D.W.); 2Molecular Pathology Laboratory, Institute of Molecular Biosciences, Mahidol University, 25/25 Phuttamonthon Sai 4, Salaya, Nakorn Pathom 73170, Thailand; wipapornkhan@gmail.com; 3Proteomics Research Laboratory, Genome Technology Research Unit, National Center for Genetic Engineering and Biotechnology, National Science and Technology Development Agency, 113 Thailand Science Park, Phahonyothin Road, Khlong Nueng, Khlong Luang, Pathumthani 12120, Thailand; atchara@biotec.or.th (A.P.); sittiruk@biotec.or.th (S.R.)

**Keywords:** Zika virus, dengue virus, andrographolide, HSPA1A, PGK1

## Abstract

Andrographolide is a labdene diterpenoid with potential applications against a number of viruses, including the mosquito-transmitted dengue virus (DENV). In this study, we evaluated the anti-viral activity of three 14-aryloxy analogues (ZAD-1 to ZAD-3) of andrographolide against Zika virus (ZIKV) and DENV. Interestingly, one analogue, ZAD-1, showed better activity against both ZIKV and DENV than the parental andrographolide. A two-dimension (2D) proteomic analysis of human A549 cells treated with ZAD-1 compared to cells treated with andrographolide identified four differentially expressed proteins (heat shock 70 kDa protein 1 (HSPA1A), phosphoglycerate kinase 1 (PGK1), transketolase (TKT) and GTP-binding nuclear protein Ran (Ran)). Western blot analysis confirmed that ZAD-1 treatment downregulated expression of HSPA1A and upregulated expression of PGK1 as compared to andrographolide treatment. These results suggest that 14-aryloxy analogues of andrographolide have the potential for further development as anti-DENV and anti-ZIKV agents.

## 1. Introduction

The genus *Flavivirus* of the family *Flaviviridae* contains some 53 viral species [1], of which 27 are transmitted primarily by *Culex* spp. or *Aedes* (*Ae*.) spp. mosquitoes. Members of this genus, which includes yellow fever virus (YFV), Japanese encephalitis virus (JEV), West Nile virus, dengue virus (DENV) and Zika virus (ZIKV), impose a significant public health burden in many tropical and subtropical countries around the world. In addition to the well known viruses of this genus, there are a number of lesser known viral species capable of causing human disease, and with the potential to emerge [2]. While vaccination offers an effective control strategy, commercial human vaccines are only available for YFV, JEV and DENV, although the DENV vaccine is not considered suitable for administration to DENV-naïve individuals [3]. Lack of universal vaccine coverage in areas of transmission, coupled with the lack of a commercial vaccine for the majority of mosquito-transmitted flaviviruses, necessitates the development of effective treatment options, but currently there is no approved drug to treat any flaviviral infection. Thus, the search for effective drugs to treat infections with these viruses is a high priority.

All members of the genus *Flavivirus* are enveloped viruses, with an approximately 11 kb positive-sense single-strand RNA genome that encodes for a single polypeptide that is cleaved by host and virally encoded proteases into 3 structural and 7 non-structural (NS) proteins [4]. The NS proteins form the replication machinery, as well as modulate the host cell proteome to favor viral replication [5] as well as dampen innate immunity [6].

Two of the members of the genus *Flavivirus* with a particularly wide distribution are DENV and ZIKV. Both were originally isolated in the 1940s [7,8], and both are predominantly transmitted by *Ae*. spp. mosquitoes, particularly *Ae*. *aegypti* and *Ae*. *Albopictus* [9]. Infection with either virus can be asymptomatic, or infection can result in relatively self-limiting symptoms including fever, rash, headache and joint and muscle pain [10]. In some cases, more severe symptoms can occur, including hemorrhagic manifestations in DENV infection [11] and neurological complications in ZIKV infection [12]. DENV has been widely distributed in tropical and sub-tropical countries for several decades [13], while ZIKV was explosively disseminated throughout much of the world over the last 5 years [14].

*Andrographis paniculata* [Burm. F.] Nees, an herb known as a “natural antibiotic”, is commonly used in China, India and Southeast Asia for the treatment of a large variety of illnesses including fever, inflammation and infection [15]. Andrographolide (Scheme 1), a bicyclic diterpenoid lactone, is one of the major components isolated from *A. paniculata* and andrographolide has been shown to have activity against a variety of viruses, including both RNA [16,17,18,19,20] and DNA [21,22,23] viruses. In particular, andrographolide has been shown to have good antiviral activity against DENV, with a half maximal effective concentration (EC_50_) of 21 to 22 μM, depending upon cell type [18]. However, andrographolide has relatively poor bioavailability [24], and efforts have been made to improve its physiochemical properties and pharmaceutical features [25,26,27]. As one part of our mission to modify andrographolide and discover its more potent analogues, in this study, we sought to evaluate the activity of three 14-aryloxy analogues of andrographolide, ZAD-1, ZAD-2 [28] and ZAD-3 [28] (Scheme 1) against DENV and ZIKV. Results showed that one new analogue, ZAD-1, had a better antiviral activity against both DENV and ZIKV than andrographolide, and so a proteomic analysis was undertaken to understand how the modification led to improved antiviral activity. Overall, the results suggest that 14-aryloxy analogues of andrographolide have significant potential for further development as anti-flaviviral agents.

## 2. Results

### 2.1. Synthesis of Compounds

The synthesis of analogues (Scheme 1) was conducted according to the previously reported synthetic method [28] by the introduction of 5,7-dichloro-8-hydroxyquinaldine into 14α-3,19-acetonylidene-protected andrographolide to form ZAD-1 (>98% high performance liquid chromatography (HPLC) purity) and part of the information is detailed in the Appendix A. ZAD-2 (>98% in HPLC purity) and ZAD-3 (>98% in HPLC purity) were synthesized as previously reported [28].

### 2.2. Evaluation of Viral Infectivity for Selection of Host Cells

As the previous studies [18,20] showed that the inhibitory activities against DENV and Chikungunya virus (CHIKV) were seen by post-treatment but not pre-treatment with andrographolide, experiments in this study were conducted at 24 h post-treatment with andrographolide and its analogues. Initially, five cell lines were selected to investigate as host cells for ZIKV infection, namely CHME-5 (a human microglial cell line), Hep3B (a human hepatocellular carcinoma cell line), BHK-21 (a baby hamster kidney cell line), Vero (an African green monkey kidney epithelial cell line) and A549 (a non-small cell lung cancer cell line) cells. These cells were mock-infected or infected with ZIKV at MOIs (multiplicity of infections) of 1, 5, 10 and 20. After 24 h of infection, the cells were collected to assess of the degree of infection by flow cytometry. The results showed that infection of CHME-5 (Appendix AA) and Hep3B (Appendix AB) resulted in low infectivity, while there was higher infectivity in ZIKV-infected BHK-21 (Figure 1A), Vero (Figure 1B) and A549 (Figure 1C) cells, which are suitable host cells for further anti-ZIKV drug screening. For DENV infection, HEK293T/17 (human embryonic kidney cell line) cells were infected with DENV serotype 2 (DENV 2) at different MOIs and the most appropriate level of infection was observed with MOI 5 (Figure 1D).

### 2.3. Evaluation of Cytotoxicity of Compounds

Cytotoxicity of each compound towards cells for four selected cell lines in the range of 1 to 500 µM was evaluated using the MTT (3-(4,5-dimethylthiazol-2-yl)-2,5-diphenyltetrazolium bromide) assay at 24 h post-treatment, in parallel with control and vehicle-treated cells (0.1% dimethyl sulfoxide (DMSO)). The 50% cytotoxic concentration (CC_50_) values calculated from the dose-response curve (Appendix A) are shown in Table 1. Overall, a range of cytotoxicities were seen from the four compounds across the five cell lines. The most cytotoxic combination was ZAD-1 and BHK-21 cells, while the least cytotoxic combination was andrographolide and BHK-21 cells (Table 1).

To determine if the compounds affected the morphology of the cells, cells from all four cell lines (A549, BHK-21, Vero and HEK293T/17) were treated with the compounds or vehicle for 24 h, and then examined under an inverted microscope (Figure 2 and Appendix A). Clear morphological changes were observed at higher treatment concentrations, but there were no obvious morphological alterations to the cells at concentrations at 10 μM or lower of each compound, with the exceptions of Vero and HEK293T/17, which showed some morphological changes at concentrations below 10 μM.

### 2.4. Screening of Compounds in ZIKV and DENV Infection

For a preliminary screening of the effect of the compounds on ZIKV infection, BHK-21, Vero, and A549 cells were infected with ZIKV at MOI of 1, 5, or 10 for 2 h. Subsequently, the cells were incubated for 24 h in the presence of the compound or vehicle and then collected, and the percentage of infection was determined by flow cytometry. The results (Appendix A) showed that there was considerable variation in the inhibition of ZIKV infection by andrographolide and the andrographolide analogues (ZAD-1, ZAD-2 and ZAD-3), with the greatest inhibition being seen in A549 cells at MOI of 5. Additionally, it is noted that in some cases, the treatment of the compounds in BHK-21 (Appendix A) and Vero (Appendix A) resulted in an increase in the level of infection, but this was not reflected by an increase in viral production. The mechanism behind this remains unclear.

To confirm the effects of andrographolide and the andrographolide analogues on ZIKV infection, A549 cells were infected with ZIKV at MOI of 5 for 2 h, and subsequently treated with the compounds at concentrations of 10 and 25 μM or with vehicle for 24 h. The cells were harvested for determination of the percentage of infection by flow cytometry, while the supernatants were collected for determination of viral titer by standard plaque assay. The results (Figure 3A) showed that ZAD-1 and andrographolide at both 10 and 25 µM significantly reduced the number of ZIKV-infected cells, but ZAD-2 and ZAD-3 showed no significant effects on ZIKV infection. The plaque assay (Figure 3B) showed that both concentrations of all compounds significantly reduced viral production. The greatest inhibitory effect on ZIKV infection was seen at 25 µM of ZAD-1.

The effects of the compounds on DENV-2 infection were evaluated using HEK293T/17 as host cells at MOI of 5. The cells were infected with DENV 2 for 2 h and then the cells were incubated with the compounds at non-cytotoxic (or morphology altering) concentrations of 10 and 25 μM of ZAD-1 and ZAD-3, or 25 and 50 μM of ZAD-2, or 50 and 75 μM of andrographolide or 0.1% DMSO vehicle control. After 24 h of treatment, cells were harvested for determination of the percentage of infection by flow cytometry, while the culture supernatants were collected to determine the viral production. The flow cytometry results (Figure 3C) showed that every compound at all tested concentrations significantly reduced the percentage of DENV 2 infection, with the exception of 10 µM of ZAD-3. The results from the standard plaque assay (Figure 3D) showed significant reductions in viral production of each compound at both concentrations. The greatest reduction was observed with 25 µM of ZAD-1, which is consistent with the results observed in the ZIKV experiments.

### 2.5. Determination of EC_50_ of ZAD-1 and Andrographolide in DENV 2 and ZIKV Infection

From the above results, ZAD-1 is the most effective compound among 3 andrographolide analogues, with a greater inhibitory effect on both ZIKV and DENV than andrographolide. Thus, the EC_50_ values of ZAD-1 and andrographolide were determined against ZIKV and DENV 2. For ZIKV, A549 cells were infected with ZIKV at MOI 5 and after 2 h of infection, the cells were incubated with ZAD-1 or andrographolide at a range of concentrations (1 to 100 µM) for 24 h, after which the supernatants were collected for determination of EC_50_ values by standard plaque assay. Both ZAD-1 and andrographolide reduced viral production in a dose-dependent manner (Figure 4A). The EC_50_ values against ZIKV infection were 27.9 ± 1.7 and 31.8 ± 3.5 µM for ZAD-1 and andrographolide, respectively. For DENV 2, HEK293T/17 cells were infected with DENV 2 at MOI 5 for 2 h and followed by incubation with ZAD-1 or andrographolide at a range of concentrations (1 to 100 µM) for 24 h. The results from the analysis of DENV 2-infected supernatants by standard plaque assay showed that treatment with both ZAD-1 and andrographolide resulted in dose-dependent reductions of viral production (Figure 4B). The EC_50_ values against DENV 2 infection were 22.6 ± 1.8 and 35.2 ± 2.5 µM for ZAD-1 and andrographolide, respectively. The EC_50_ value for andrographolide in DENV infection is consistent with our previous study undertaken in HepG2 and HeLa cells with the same virus [18]. Determination of the selectivity index for ZAD-1 and andrographolide for both ZIKV and DENV showed that the compounds were having specific antiviral effects over and above and effects of cytotoxicity (Table 2). The highest selectivity index was seen with ZAD-1 towards ZIKV (Table 2).

### 2.6. Differential Expression of Host Cell Proteins after Treatment with ZAD-1 and Andrographolide

To investigate why ZAD-1 showed better anti-viral activity as compared to andrographolide, a proteomic analysis was undertaken. A549 cells were therefore treated with either vehicle (0.025% DMSO), or with 25 μM of andrographolide or ZAD-1, and at 24 h post-treatment, proteins were prepared and subjected to two-dimension polyacrylamide gel electrophoresis (2D-PAGE). The experiment was undertaken independently in triplicate and representative gels are shown Figure 5, with all replicates being shown in Appendix A. After image analysis, 8 significantly (*p* < 0.05) differentially expressed protein spots were identified. All of these protein spots were cut from the gels and then subjected to in-gel tryptic digestion followed by mass spectrometry. MASCOT (Matrix Science Inc., Boston, MA, USA) search of the resultant masses identified 4 proteins (Table 3), namely heat shock protein 70 kDa 1A (HSPA1A), phosphoglycerate kinase 1 (PGK1), transketolase (TKT) and nuclear transport protein of GTP-binding nuclear protein Ran (Ran).

### 2.7. Validation of Differentially Expressed Proteins

To confirm the 2D-gel analysis results, HSPA1A and PGK1 were selected for the validation by Western blot. The same three experimental conditions were established, namely A549 cells treated with vehicle (0.025% DMSO), 25 μM andrographolide or 25 μM ZAD-1. Glyceraldehyde-3-phosphate dehydrogenase (GAPDH) was used as a loading control and the proteins were normalized to this protein. Results showed that 25 µM andrographolide significantly increased HSPA1A expression compared with the vehicle control, while ZAD-1 treatment showed no significant difference from control (Figure 6A). While the difference in HSPA1A expression between andrographolide treatment and ZAD-1 treatment is consistent with the spot intensity data (Table 3), the spot density for the control DMSO treatment is not consistent with the western blot result. PGK1 expression in 25 µM andrographolide-treated cells was significantly upregulated relative to the vehicle, and ZAD-1 treatment resulted in significantly increased expression as compared to andrographolide treatment 

## 3. Discussion

The mosquito-transmitted members of the genus *Flavivirus* impose a significant public health burden around the world. As was shown by the recent spread of Zika virus around the world [14], the presence of competent mosquito populations in many parts of the world and modern travel can allow the rapid spread of an emerging pathogen across a significant portion of the globe. Despite decades of study, the number of available vaccines to protect against the different flaviviruses remains small, and there are no specific drugs for treatment available.

Andrographolide, a bicyclic diterpenoid lactone, is a major bioactive constituent of *Andrographis paniculata*, and has been shown to have activity against a number of viruses including DENV [16,17,18,19,20,21,22,23,29,30,31]. Given the close relationship between DENV and ZIKV, which belong to the same genus, it seems reasonable to expect that andrographolide would have antiviral activity against ZIKV, as was shown here. In addition to andrographolide, three andrographolide analogues were also evaluated, one of which, ZAD-1, with a 14-(8′-quinolyloxy) group, had improved activity against both ZIKV and DENV as compared with the parental compound, while the remaining two compounds (ZAD-2 and ZAD-3) with 14-phenoxy groups were less active than the parental compound. This suggests that specific heteroaryloxy (e.g., quinolyloxy) analogues of andrographolide have the potential for further development for improved antiviral activity.

We note that the EC_50_ of andrographolide towards DENV 2 in this study (35.2 μM) determined in HEK293T/17 cells was somewhat higher than the previously reported values of 21.3 and 22.7 μM, as determined in HepG2 and HeLa cells respectively [30], suggesting that host cells play a role in the efficacy of andrographolide. Host cell type specificity in the response of cells towards andrographolide has been previously reported [32,33].

To gain some insights into the mechanism by which ZAD-1 has improved activity over andrographolide, a 2D proteomic analysis was undertaken. A total of eight differentially expressed spots were identified, from which four proteins (HSPA1A, PGK1, TKT, Ran) were identified by mass spectrometry. In a previous proteomic analysis of the mechanism of action of andrographolide [31], 17 proteins were identified as differentially expressed in response to andrographolide treatment of HepG2 cells. In this study, four proteins of a chaperone of HSPA1A, two enzyme proteins of PGK1 and TKT and a nuclear protein of Ran were revealed to have expression differences by the actions of andrographolide and ZAD-1, indicating the different working consequences between andrographolide and ZAD-1. On the other hand, only four proteins’ expressions appear altered, illustrating that there is consistency between the actions of two compounds and that most features of them from the same structural backgrounds are shared. The two proteins evaluated for their response to andrographolide both showed significant differences between the two treatments, although we note that the DMSO control for HSPA1A is not consistent between the Western blot and the original 2D proteome spot intensity.

The four identified proteins are of diverse biological functions, with HSPA1A being a stress-induced chaperone protein of the Hsp70 family of proteins [34], PGK1 is an enzyme that is involved in glycolysis, although it may have other functions [35,36], TKT is an enzyme of the pentose phosphate pathway [37], while Ran is a nuclear translocation protein [38]. Hsp70 and Ran were identified to interact with JEV NS5, and knockdown of Hsp70 by the short-hairpin RNA (shRNA) corresponding to the HSPA1A mRNA sequences resulted in a significantly reduced JEV genome replication [39]. It is reported in vitro that DENV activates hypoxia inducible factor (HIF) and anaerobic glycolysis markers and enhanced DENV RNA replication correlates directly with an increase in anaerobic glycolysis, producing elevated adenosine triphosphate (ATP levels) [40]. As PGK1 and TKT are involved in energy metabolism, it is worth noting that TKT links the pentose phosphate pathway and the glycolysis pathway and active glycolysis has been shown to be required for DENV infection [41]. Thus, even small perturbations of these pathways could have a large effect on viral replication. At this point, it is unclear if the enhanced antiviral activity of ZAD-1 is a consequence of the effect of one protein or all four proteins (or any combination thereof).

The analysis used here to gain more information on the improved antiviral activity of ZAD-1 as compared to andrographolide, 2D proteomic analysis, while having relatively high resolution, is not a particularly sensitive proteomic technique, with a limit of spot sensitivity of approximately 10 ng [42]. Thus, it is likely that a more sensitive technique would show a larger number of proteins differentially expressed after treatment with ZAD-1 or andrographolide. The advantage of the 2D methodology is that it is relatively robust, and generally, proteins identified as differentially regulated can be validated. From the 2 proteins evaluated, only one DMSO control was discordant between the proteomic analysis and the Western blot analysis.

Andrographolide has been shown to have antiviral activity against a number of mosquito-transmitted positive-sense RNA viruses, including DENV [18,19,31], CHIKV [20] and now ZIKV. However, there remains significant uncertainty as to how the antiviral effect of andrographolide is mediated. Studies have shown that andrographolide can modulate a number of cellular processes, including autophagy [21,43], the response to cellular [44,45] and oxidative [46] stress, as well as mitochondrial functions [47]. All of these pathways can directly impact on viral replication. Other possible mechanisms include andrographolide exerting its antiviral effect through heme oxygenase I [19,48], nuclear factor kappa-light-chain enhancer of activated B cells (NF-kB) [48] or glucose regulated protein 78 (GRP78) [30]. However, this and other studies have shown that andrographolide has good broad-spectrum antiviral activities, and this study has shown that those activities can be further improved.

In conclusion, this study revealed that there are different working consequences between andrographolide and ZXD-1 bearing quinolyloxy even though most of the pharmaceutical features of andrographolide and ZAD-1 from the same structural backbone are shared. The four identified proteins of HSPA1A, PGK1, TKT and Ran by 2D gel proteomic analysis may play important roles in anti-DENV and anti-ZIKV activity. Therefore, these results suggest that 14-aryloxy analogues of andrographolide have potential for further development as anti-DENV and anti-ZIKV agents, and this work provides one way to discover potent inhibitors against ZIKV and DENV from natural products.

## 4. Materials and Methods

### 4.1. Chemicals and the Stock Solutions

Andrographolide was purchased from Sigma-Aldrich (St. Louis, MO, USA). Andrographolide as well as ZAD-1, ZAD-2 or ZAD-3 were dissolved in 100% DMSO to final stock concentrations of 100 mM and stored at −80 °C. Each compound was serially diluted to various concentrations using complete Dulbecco’s modified Eagle media (DMEM). The final concentration of DMSO in media was set to 0.1% for the drug screening, or 0.025% for the 2D-PAGE electrophoresis and Western blot.

### 4.2. Synthesis of ZAD-1

Under N_2_ atmosphere and at 0 °C, 0.5 g (1.28 mmol) 3,19-acetonylideneandrographolide, 5,7-dichloro-8-hydroxyquinaldine (1.54 mmol) and 0.5 g PPh_3_ (1.92 mmol) were dissolved in 20.0 mL anhydrous THF. To the above mixture, the solution of 0.39 g (1.92 mmol) DIAD in 2.0 mL anhydrous THF was added dropwise in 5 min and the reaction progress was monitored by thin layer chromatography (TLC). After the reaction was complete in about 2 h, the reaction mixture was treated with ethyl acetate and saturated NaHCO_3_ solution. The organic phase was washed with brine and dried over anhydrous Na_2_SO_4_. Filtered, dried and silica gel chromatographed by petroleum ether and ethyl acetate to yield 590 mg (76.7% yield) of titled product ZAD-1; m.p. 155–157 °C; ^1^H NMR (400 MHz, DMSO-*d*_6_) *δ* 8.47 (d, *J* = 8.7 Hz, 1H), 7.93 (s, 1H), 7.69 (d, *J* = 8.7 Hz, 1H), 6.61 (dd, *J* = 8.5, 4.8 Hz, 1H), 6.47 (d, *J* = 4.5 Hz, 1H), 4.74–4.67 (m, 2H), 4.60 (dd, *J* = 11.0, 4.7 Hz, 1H), 4.21 (s, 1H), 3.75 (d, *J* = 11.6 Hz, 1H), 3.28 (dd, *J* = 9.5, 4.3 Hz, 1H), 3.03 (d, *J* = 11.6 Hz, 1H), 2.75 (s, 3H), 2.23 (d, *J* = 11.8 Hz, 1H), 1.83 (m, 2H), 1.73 (m, 1H), 1.66 (d, *J* = 10.9 Hz, 1H), 1.58–1.52 (m, 2H), 1.47–1.38 (m, 1H), 1.30 (s, 3H), 1.23 (s, 3H), 1.08 (m, 5H), 0.90 (m, *J* = 12.1, 5.7 Hz, 1H), 0.74 (m, 1H), 0.43 (s, 3H) ppm; ^13^C NMR (101 MHz, DMSO) *δ* 169.5, 160.4, 149.1, 147.5, 146.7, 142.5, 133.3, 127.0, 126.6, 126.2, 126.0, 124.2, 124.1, 108.3, 98.1, 76.8, 75.8, 71.8, 62.6, 54.4, 51.4, 37.7, 37.0, 36.9, 33.5, 27.7, 25.8, 25.4, 25.2, 25.0, 24.9, 22.6, 15.2 ppm; HRMS (ESI) *m/z* 600.2274 [M + H]^+^, calculated for C_33_H_40_Cl_2_NO_5_, 600.2284; *m/z* 602.2256 [M(^37^Cl)+H]^+^, calculated for C_33_H_40_Cl^37^ClNO_5_, 602.2254; *m/z* 604.2266 [M(^37^Cl_2_) + H]^+^, calculated for C_33_H_40_^37^Cl_2_NO_5_, 604.2225.

### 4.3. Cell Lines and Viruses

The human lung cell line A549 (ATCC Cat No. CCL-185), the human microglial cell line CHME-5 [49] the human hepatoma cell line Hep3B (ATCC Cat No. HB-8064) and the human embryonic kidney cell line HEK293T/17 (ATCC Cat No. CRL-11268) were cultured in Dulbecco’s modified Eagle’s medium (DMEM) supplemented with 10% heat-inactivated fetal bovine serum (FBS), 100 units of penicillin and 100 mg streptomycin/mL at 37 °C with 5% CO_2_. The monkey kidney cell line Vero (ATCC Cat. No. CCL-81) was cultured under the same conditions but supplemented with 5% FBS. The baby hamster kidney cell line BHK-21 (ATCC Cat No. CCL-10) was cultured in RPMI-1640 medium supplemented with 10% fetal calf serum (FCS) at 37 °C with 5% CO_2_. The *Ae*. *albopictus* cell line C6/36 (ATCC CRL-1660) was cultured at 28 °C in MEM (Gibco, Invitrogen) supplemented with 10% FBS and 100 units of penicillin and 100 mg streptomycin/mL. Cell lines were selected based upon our previous studies [50,51,52].

DENV 2 (strain 16681) and ZIKV (strain SV0010/15) virus stocks were propagated in C6/36 cells previously described [53], and virus titers were determined by standard plaque assay as described previously [52,54] using Vero cells for ZIKV and LLC-MK_2_ cells for DENV.

### 4.4. Cell Viability Assays

A549 cells were cultured in ninety-six-well tissue culture plates under standard conditions until the cells reached 90% confluence. The cell culture medium was removed and then cells were incubated with various concentrations of andrographolide (1–100 μM) diluted in complete medium with FBS prior to being cultured for 24 h under standard conditions before analysis using either the Vybrant MTT Cell proliferation assay kit (V13154, Invitrogen) or the alamarBlue (Thermo Fischer Scientific, Waltham, MA, USA) assay kit according to the manufacturers recommendations. Values were determined from 8 (MTT assay) or 3 (alamarBlue) independent replicates. Negative controls (only media and media with 0.1% DMSO) and positive controls (5% DMSO) were included. To determine cell number, cell cultures were trypsinized by the addition of 0.25% Trypsin/0.1% ethylelediaminetetraacetic acid (EDTA) for 3 min and then cell suspensions were diluted with 0.4% trypan blue solution. Cells were counted using a hemocytometer. Cell morphology was additionally observed under an inverted microscope after 24 h treatment with various concentrations of each compound.

### 4.5. Effects of Compounds on Viral Infection

HEK293T/17 or A549 cells were cultured in six-well plates under standard conditions until the cells reached 80% confluence. HEK293T/17 cells were infected with DENV 2, while A549 cells were infected with ZIKV Thai strain at MOI 5 for 2 h at 37 °C and 5% CO_2_ in culture medium without FBS. Mock infection was undertaken in parallel. After the infection period, the medium was removed and replaced with complete medium (including FBS) containing 0.1% DMSO (no treatment control) or the appropriate concentration of each compound and cells were again incubated under standard conditions. At 24 h post-treatment, cells were harvested for determination of effects of compounds on percentage of infection and supernatant was collected for determination of viral productivity. All experiments were undertaken independently in triplicate with duplicate plaque assay.

### 4.6. Flow Cytometry

Mock- or viral-infected cells (with or without compound treatment) were harvested at an appropriate time point then incubated with 10% normal goat serum (NGS; Gibco Invitrogen) in PBS-immunofluorescence assay (IFA) buffer on ice for 30 min. The cells were washed with 800 µL of 1% BSA/PBS-IFA and fixed with 200 µL of 4% paraformaldehyde in PBS-IFA at room temperature in the dark for 20 min. After washing with 800 µL of 1% BSA/PBS-IFA, cells were permeabilized with 200 µL of 0.2% Triton X-100 in PBS-IFA for 10 min at room temperature in the dark. Subsequently, the cells were washed with 800 µL of 1% BSA/PBS-IFA followed by overnight incubation with 50 µL of primary antibody (1:150 dilution of pan-specific anti-dengue virus E protein antibody HB114 for DENV and 1:2 dilution of anti-flavivirus antibody HB112 for ZIKV). After washing twice with 800 µL of 1% BSA/PBS-IFA, the cells were incubated with 50 µL of a goat anti-mouse IgG conjugated with fluorescein isothiocyanate (FITC; KPL, Guilford, UK) diluted 1:40 with 1% BSA/PBS-IFA at room temperature in the dark for 1 h. The cells were washed twice with 800 µL of 1% BSA/PBS-IFA and resuspended in 100 µL of PBS-IFA. The fluorescence signal was analyzed by flow cytometry on a BD FACalibur cytometer (Becton Dickinson, BD Biosciences, San Jose, CA, USA) using CellQuest™ software (Version 6). All experiments were undertaken independently in triplicate. All raw flow cytometry data plots are provided in Appendix A.

### 4.7. 2D-PAGE and Liquid ChromatographyTandem Mass Spectrometry (LC-MS/MS) Analysis

2D-PAGE and mass spectroscopic identification of differentially expressed spots was undertaken essentially as previously described [30]. Briefly, A549 cells treated with vehicle, andrographolide or ZAD-1 for 24 h were harvested and proteins prepared, and a total of 250 µg were loaded onto Immobiline Drystrips (pH 3–10, NL, 7 cm) for isoelectric focusing using Ettan IPGphore II system (Amersham Biosciences Amersham, UK). The second dimension electrophoresis was undertaken on 12.5% polyacrylamide gels. Gels were stained with 0.1% colloidal Coomassie Brilliant Blue G 250 with 20% methanol for 24 h and destained with milli-Q water for 48 h. The resultant gels were scanned under visible light at 300 µm/pixel resolution. Image data were analyzed using Image Master™ 2D Platinum software version 7.0 (GenBio, Geneva, Switzerland). Each spot was evaluated on intensity, volume and area. Statistical analysis was performed by one-way analysis of variance (ANOVA), with a *p*-value of 0.05 being considered significant. The experiment was performed as three independent biological replicates for each condition. After tryptic in-gel digestion, peptide masses were determined by LC-MS/MS using SYNAPT high definition MS (HDMS) (Waters Corp., Milford, MA, USA).

### 4.8. Western Blotting

A total of 40 µg of proteins were separated by electrophoresis through a 12% sodium dodecyl sulfate (SDS) polyacrylamide gel and transferred to a nitrocellulose membrane by electroblotting. The membranes were blocked with 5% skim milk in TBS-T (20 mM Tris, 140 mM NaCl and 0.1% Tween-20) at room temperature for 1 h with shaking. The membranes were probed overnight with 1:1000 dilution of a rabbit polyclonal anti-heat shock 70 kDa 1A (HSPA1A) antibody (AV33096; Sigma-Aldrich), or 1:5000 dilution of a rabbit polyclonal anti-phosphoglycerate kinase1 (PGK1) antibody (PA5-28612; Thermo Scientific™, Waltham, MA, USA) as primary antibodies at 4 °C. After washing three times with TBS-T for 5 min each time, the membranes were then incubated with 1:2000 (for anti-HSPA1A antibody), or 1:5000 (for anti-PGK1 antibody) dilution of a horseradish peroxidase (HRP)-conjugated polyclonal goat anti-rabbit IgG antibody (31460; Thermo Scientific) as a secondary antibody at room temperature for 1 h. The signal of specific protein bands was detected using the Luminata Forte Western HRP Substrate (Millipore Corp., Burlington, MA, USA) and recorded by film autoradiography. All experiments were undertaken independently in triplicate.

### 4.9. Statistical Analyses

All data were analyzed using the GraphPad Prism program (GraphPad Software, version 5.01). Statistical analysis of significance was undertaken by ANOVA with Tukey’ post hoc comparisons on raw data reads using SPSS, *p* < 0.05 for significance (SPSS Inc., Chicago, IL, USA). CC_50_ and EC_50_ values were calculated using the freeware ED50plus (v1.0) software (http://sciencegateway.org/protocols/cellbio/drug/data/ ed50v10.xls).

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
