# Peer review of "Andrographolide and Its 14-Aryloxy Analogues Inhibit Zika and Dengue Virus Infection"

_molecules, 2020, doi:10.3390/molecules25215037_

Round 1
Reviewer 1 Report
the publication concerns the evaluation of antiviral activities (ZIKA and Dengue) of andrographolide and aryloxy derivatives with the aim of changing the physico-chemical properties and therefore the bioavailability. After determining the infected host cells relevant to the study, the authors determined the EC50 and cyctotoxicity.
The authors also describe a 2D proteomic analysis that allows them to identify targets affected by during the viral infection.
However, reading the publication leads to a few questions:
1- it has been shown that andrographolide has antitumor activity on A-549 cells (doi.org/10.1016/j.ejphar.2010.01.009). The decrease in infection in the presence of Andro or ZAD-1 could be explained by an antitumor activity. Have tests been carried out to rule out this possibility?
2 - the choice of cell lines: the cell lines chosen are not referenced as being host cells for Zika (doi: 10.3390 / ijms20051101). Could the authors explain their choice of cells?
3- concerning the supporting information and the NMR spectrum of compound ZAD-1: the aryl ring signals on the spectrum look like a para-disubstituted aromatic signal, which is not the case in ZAD-1. Spectrum could be checked
Author Response
Thanks for reviewing “molecules-940379”!

Reviewer 2 Report
The manuscript entitled " Andrographolide and Its 14-Aryloxy Analogues Inhibit Zika and Dengue Virus Infection" reports a study of anti-viral activity of three 14-aryloxy analogs of andrographolide against dengue zika virus. There are some points to change.
1- Abstract
"dengue virus (DENV). In this study, we 18 evaluated the anti-viral activity of three 14-aryloxy analogs (ZAD-1 to ZAD-3) of andrographolide against Zika virus"
Please replace "Zika" with zika"
2 -Please, add the values of EC50 values against ZIKV and DENV 2 in a table and compare with the values of table 1, and add the selectivity index. This will improve the discussion regarding the differences in the biological activity of the compounds.
3 - Please, I propose that the authors propose the mechanism of action using some in silico analysis, for example, docking.
Author Response

(The authors gave the same response as above.)

Reviewer 3 Report
Andrographolide and Its 14-Aryloxy Analogues 2 Inhibit Zika and Dengue Virus Infection 3 Feng Li1,†, Wipaporn Khanom2,†, Xia Sun1, Atchara Paemanee3, Sittiruk Roytrakul3, Decai Wang1, 4 Duncan R. Smith2,* and Guo-Chun Zhou1,*
The authors tested Andrographolide and three new derivatives for antiviral activity against ZIKV and DENV infection in vitro. The analogue with the best antiviral activity was chosen for cell proteomic analysis which identified four, differentially expressed proteins.
Figure 1. The percent infection appears low for these cells, especially Vero cells at 20 MOI. The methods refer to another paper for the “standard plaque assay” used to titer the virus stocks. Please indicate in the methods section what cell type was used for the plaque assay, and further, describe the methods for infections of test cells, ie, was virus bound at cold temperature prior to a shift for entry or was virus added to the cultures at 37°. Was the virus inoculum removed from the cells after a binding period or left on for 24 hrs?
Please provide supplemental figures with the flow cytometry data, so that the reader can evaluate the sensitivity and background levels of the flow assay.
Table 1 and supplemental figure 2-5
The text interprets the results of cytotoxicity testing as “Overall, no obvious cytotoxicity was seen at 24 hrs post treatment for each compound.” That does not appear to be the case as there was significant toxicity for all compounds. The statement should be modified to indicate the concentration ranges that were non-toxic and were presumably the reason for moving forward with concentrations lower than 25µM. ZAD-1 appears to have significant toxicity in HEKs at concentrations as low as 10µM (figure S5).
The label for the Y-axis is cut off in figure S2.
Figure 2 and S6-S8
The text states “there were no obvious morphological alterations to the cells at concentrations of lower than 25 μM of each compound.” Figure 2 shows morphological alterations in A549 cells at 25µM concentrations. Figure S8 shows morphological alterations in Veros at 10µM concentrations of ZAD-1, 2, and 3, and ZAD-1 shows alteration of HEK morphology at 10 µM ZAD-1 corresponding to its apparent toxicity in these cells (Figure S5). The text should read “at 10µM and lower” and further indicate that the specific cases for Vero and HEKs at 1 µM or lower with the ZAD analogues.
Figures 3 and S9-11
“…considerable variation in the inhibition of ZIKV infection…”is an interesting interpretation of the data, when the drugs enhanced infection in BHKs (figure S9). Were those increases significant? Do the statistics in these charts represent values for all three biological replicates or single replicates in a single flow analysis? In figure 2, the most significant reductions in positive cells and PFUs correspond with the data indicating cytopathicity with ZAD-1 in A549 and HEK cells. Did the flow cytometry identify apoptotic cells by exclusion of propidium iodide or staining with markers of apoptotic proteins?
Figure 4. The EC50 data largely matches the cytotoxicity and cell morphology data.
Figures 5 and 6 and table 2.
Was HSPA1A increased or decreased by either drug treatment?
Author Response

(The authors gave the same response as above.)

Round 2
Reviewer 3 Report
The authors have addressed the major concerns.
Please report the statistical values for all of the data in Figure S9, panel A.
Author Response
Response: We thank the reviewer for persisting with this point, and indeed the reviewer is correct that we only did the statistical analysis for a reduction in infection. We have now added the statistical analysis for Figure S9 as well as Figure S10 and all statistical significance has been indicated. In addition, we have commented upon this in the manuscript text (see lines 135 to 137).